# LEARNING TO SELECT:
# PROBLEM, SOLUTION, AND APPLICATIONS

## ABSTRACT

We propose a "Learning to Select" problem that selects the best among the flexible size candidates. This makes decisions based not only on the properties of the candidate, but also on the environment in which they belong to. For example, job dispatching in the manufacturing factory is a typical "Learning to Select" problem. We propose Variable-Length CNN which combines the classification power using hidden features from CNN and the idea of flexible input from Learning to Rank algorithms. This not only can handles flexible candidates using Dynamic Computation Graph, but also is computationally efficient because it only builds a network with the necessary sizes to fit the situation. We applied the algorithm to the job dispatching problem which uses the dispatching log data obtained from the virtual fine-tuned factory. Our proposed algorithm shows considerably better performance than other comparable algorithms.

## 1 INTRODUCTION AND MOTIVATION

In a very complex manufacturing factory such as semiconductor fab, each equipment should select the next job among the candidate jobs waiting for the equipment. This kind of selection is usually called "job dispatching", and is typically done by heuristic rules. As dispatching rules are not perfect, next job selection is often overridden by human experts decision. The motivation of this research is to devise a method for learning dispatching rule from the job dispatching log in supervised learning manner. When seen from supervised learning point of view, job dispatching is different from typical classification or regression in that job selection is a relative decision. In other words, job selection is to be made by considering the entire candidate jobs. Attributes (features) of a candidate job do not provide enough information for selection. When selecting a job to work on next, the situation of the factory should be considered as well as the attributes of jobs. And the number of candidate jobs varies each time. Hence, at each moment of decision, the entire set of candidate jobs combined with the factory situation form an instance of "selection problem". We will call this problem as "Learning to Select (LtoS)". This is similar to "Learning to Rank (LtoR)" problem, however, LtoS is a special case of LtoR in that we are interested in only the "best" candidate, instead of full ranking of candidates.

In the next chapter, we define the LtoS problem and distinguish it from existing problems, and then propose Variable-Length Convolutional Neural Network, a new method to solve LtoS problem. It is applied to the dispatching problem in the factory and confirmed that it shows much better results than the existing methods. We conclude with the summary and the future research of this study.

## 2 PROBLEM DEFINITION

We present the "Learning to Select" problem in this paper. This problem is simply a matter of choosing the best candidate among the given candidates in a specific situation. In each situation, the number of candidates can be changed, and therefore, "Learning to Select" differs from other problems in that it selects the best candidate among the flexible candidates each time.

In this section, we give a general description of LtoS with an example of job dispatching. At each decision making point, the manufacturing factory status $\mathfrak{M} = \{m^1, m^2, ..., m^D\}$ is observable. Each superscript denotes the event that the job is selected, so $D$ is size of the whole training data. For each factory status $m^k$ has a list of candidate jobs $\mathfrak{J}^k = (j_1^k, j_2^k, ..., j_{n_k}^k)$, where $n_k$ is the size

of candidate jobs at the event $k$. At each event, $\mathfrak{J}^k$ is associated with the selected history $S$ which can be written as $S^k = \left( s_1^k, s_2^k, ..., s_{n_k}^k \right)$. Since only one job is selected at each event, single element of the selected history vector $S$ is 1 and others are all 0. we combine the manufacturing factory status $\mathfrak{M}$ and the each candidate $\mathfrak{J}$ to build a feature vector $X^k$. We define the feature pair for each candidate $X_i^k = (m^k, j_i^k)$ for $i = 1, ..., n_k$ and the feature vector as $X^k = (X_1^k, X_2^k, ..., X_{n_k}^k)$. We want to create a selection function $f$ that matches $S^k = f(X^k)$ well. So the training data is $\left\{ X^k, S^k \right\}_{k=1,...,D}$.

The name "Learning to Select" came from the well-known "Learning to Rank" problem. In the case of existing LtoR problems, the goal is to find a ranking function that sorts the given data well under certain conditions. But the problem we have is a little bit different. It is not sorting the given data well under certain conditions, but to find the data that will rank the highest among the data which is same as Top-1 ranking problem. Basically this is a special case of the listwise LtoR problem, so the problem formulation is quite similar to listwise LtoR (Cao et al., 2007). One of difference is that the existing listwise LtoR use the refined value of the query-document pair with the feature function before training, while we use the raw data itself to learn the features automatically during learning, and the another point is the speciality of the training data. The training data for LtoR is generally labeled query-document pairs, and their labels are indicated by scores ranging from 1 to 5 such as *Best match* or *Good match*. In this case, however, the label appears as *Selected* or *Not Selected*, and only one of the candidates has *Selected* label.

The LtoS problem can also be regarded as a problem of classification with two or more labels for given data. However, it is not a general classification problem because it is necessary to classify each subset data from whole data given by each situation, rather than classifying all possible data into two labels. Thus, even in the case of completely identical candidates, the label can be 1 in some situations, and in some other situations the label can be 0. Therefore, this problem seems similar to the classification problem, but it is slightly different.

Therefore, this study defines a LtoS problem and presents a way to solve this problem much better than existing methods. This approach combines the deep learning, which is very prominent in the existing classification problem, and the advantage from LtoR, which is the idea that considers only the subset of documents corresponding to the query.

## 3 PROPOSED APPROACH

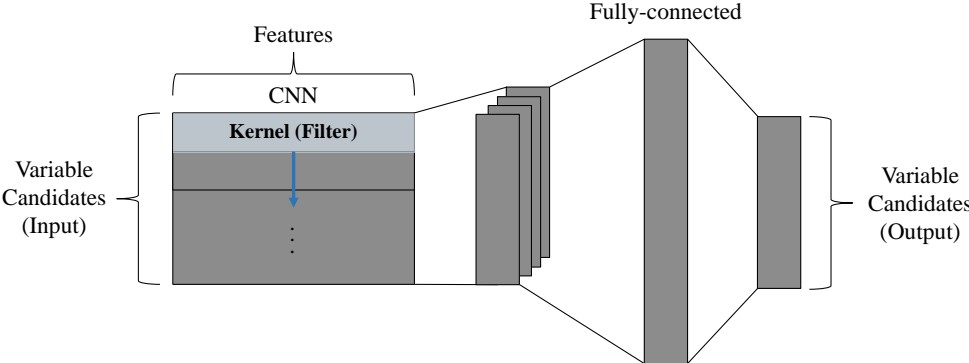

Figure 1: Structure of VLCNN

For the problem of classifying data into several classes, neural networks have dominated with high performance. However, the neural networks are not an adequate answer for every problem. In the case of this selection problem mentioned above, it seems quite similar to the classification problem, but it is a completely different problem. Therefore, the neural network for general classification problem does not solve the selection problem well. For this reason, an algorithm specific to the LtoS is needed, which can be motivated from LtoR algorithm in which chooses the most appropriate among the possible candidates each time. The most common method of LtoR is RankSVM (Joachims, 2002; Radlinski & Joachims, 2005; Kuo et al., 2014), which learns the relationship between candidates irrespective of the number of candidates, so learning is possible even if the number

of candidates changes every moment. However, it is difficult to handle with a lot of data due to the nature of SVM, and it is difficult to apply it to complex and data-intensive selection problems, because the kernel must be tuned to fit complex functions. Thus RankNet, which can learn the relationship between candidates and fitting complicated functions, has been proposed (Burges et al., 2005; Liu et al., 2009). It is a structure that combines a loss function and a neural network for the ranking problem. However, RankNet focuses on expressing the relationship between candidates for ranking, so it cannot cope when the number of candidates changes due to nature of neural network structure. Therefore, we propose an algorithm that is specific to LtoS problem.

The algorithm proposed in this paper can select the most suitable candidate like the classification problem, and it is applicable even when the number of candidates changes every moment as in the LtoR problem. It can also learn complex functions or selection rules using convolutional neural networks as the main structure. That is, a neural network structure is required which can change the number of inputs and outputs whenever the number of candidates changes. Similarly, in the semantic modeling of sentences which the number of inputs can vary, an architecture, Dynamic Convolutional Neural Network (DCNN), has been proposed to handle input sentences of varying length (Kalchbrenner et al., 2014). The DCNN induces a feature graph over an input sentence. In wide convolution and dynamic $k$-max pooling of DCNN, the subgraphs are computed from the full induced graph, and it causes a lot of computation waste. Therefore, DCNN is not suitable as a selection problem because it extracts features from sentences and classifies them by expressing semantic contents of sentences.

In this way, we apply Dynamic Computation Graph (DCG) (Looks et al., 2017) to use the neural network structure that can handle candidates with varying numbers to fit the LtoS problem. DCG makes it possible to apply the neural network to problems of various domains by computing when the structure and size of the neural network change for every input. In our problem, DCG can change the input and output structure of the fully used feature graph. This also reduces unnecessary computations. As shown in Figure 1, the proposed structure for solving the selection problem is called Variable Length CNN (VLCNN) by combining DCG with the above-mentioned convolutional structure.

In VLCNN, each candidate is computed by sharing the network weight of convolutional neural network, and the most appropriate candidate is selected. The kernel weight sharing of CNN plays a role of extracting common features from each candidate. In addition, by sharing parameters in fully-connected layers for features extracted from the each candidate in the convolutional layers, each candidate combines the common features in the same way to yield a score. The scores of each candidate from the fully-connected layer are the probability of being selected through soft-max function. That is, for each candidate, a score is given using a sub neural network having the same parameters, and the candidate is selected using the score. In these processes, DCG makes it possible to change the structure of the full neural network according to the number of candidates in a situation where the number of candidates changes. For this reason, this neural network becomes a neural network structure that can select one of the changing candidates. This neural network structure is suitable for the problem that the rules for selection are complex and the number of candidates changes at every moment.

## 4 EXPERIMENT

### 4.1 DATA DESCRIPTION

We use the job dispatching problem in fab for the experiment. It it not only because it has the characteristics of LtoS problem well, but also because the value of the application in practice is quite high. As we mentioned at the beginning, job dispatching means that the equipment selects the next job among the candidate jobs waiting for the equipment.

Since it is difficult to use actual fab data, we used data from a fine-tuned fab simulator. The data contains both the factory status and the list of candidate jobs. Also, the properties of each job candidates are included. The summary of data description in Table 1. The factory status has which equipment is needs dispatching, what process the equipment is currently processing, and how many job candidates are waiting for that equipment. Each job candidates also contains some information such as which process the job is waiting for, how long it is currently waiting, how much time has passed since it was entered the fab, etc. In general, the heuristic rule uses this properties to make decision with simple equation. For example, if the waiting time is longer, or if the current process

Table 1: Data Description: Properties of Factory Status and the Candidate Jobs.

| Factory Status | Candidate Jobs |
|---|---|
| Dispatching Equipment Current Process # of Waiting Jobs | Next Process Waiting Time Total Process Time Completeness Location |

of the equipment and the next process of the job are matched, the higher score is assigned to each job, and the job having the highest score is selected. However, these heuristic rules are incomplete because the manufacturing factory is very complicated, and because of this, many details are too frequently to be unexpected occurs and it is hard to defined as a rule. Therefore, it often causes the human expert to override it based on its rich experience. Therefore, the factory has a dispatching log that records the dispatching information after determined by the human expert overriding. In order to overcome this inefficient situation, we aim to learn the dispatching rule directly from the dispatching log which reflects the human expert's selection information.

An artificial log data from fine-tuned fab simulator contains all information in Table 1. Inside the simulator, a complex heuristic rule is designed to make a job selection, and when a specific situation comes, the virtual human expert overrides the decision with another rule than the one previously defined. We limited the maximum size of job candidates are 100, and the actual number of job candidates varies depends on fab situation. So size of the training data for each decision event is $(3 + 5 \times$ # of candidates $+ 1) \times 1$ as Figure 2. Selection history log is given as a single number as in end of Figure 2, but in actual training, we convert this single number to one-hot encoding.

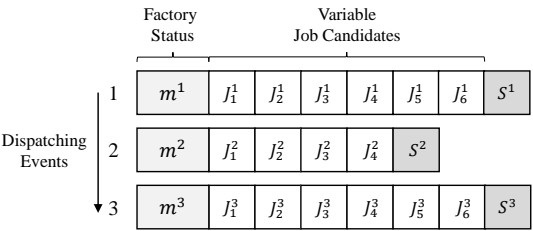

Figure 2: Structure of Sample Data

The simulator can have a maximum of 100 job candidates considering the actual physical space constraints, and the number of job candidates will vary depending on the fab situation. In this experiments, we generates 100K samples to train the network.

## 4.2 RESULTS

For the data mentioned above in 4.1 Data Description, we apply VLCNN to solve the job dispatching problem. For comparison, we apply the common neural networks, CNN and Fully-connected which are often used for existing classification. For fair accuracy analysis of each neural network structure, we make each neural network as similar as possible. In addition, all neural network structures are trained in Adam (Kingma & Ba, 2014). We also apply RankSVM, which is one of powerful LtoR algorithms, to solve the job dispatching problem as a top-1 ranking problem.

**VLCNN.** In this neural network, a convolutional layer with 128 convolutional kernels that fit the feature size and a fully connected layer with 256 hidden units are used.Since the input and output structure of the neural network changes according to the number of changing jobs, it is possible to select the best job among the candidate jobs. In addition, by sharing the weights in the neural network, the same dispatching rules are applied to each job candidates to increase the accuracy of selection.The number of parameters in this structure is a fixed number, and the number of

parameters does not change even if the number of candidate jobs increases.

**CNN.** Like the VLCNN, 128 kernels and 256 hidden units are used in the convolutional layer and the fully connected layer, respectively. Due to weight sharing of CNN, the features of each job are extracted in the same way. However, in the conventional neural network, since the computation graph is fixed before data feeding, the candidate job is fixed to the maximum number of jobs. This not only wastes computational graphs, but also reduces the accuracy of selection because all jobs in a Fab and padded parts that do not exist actually are computed. In addition, it looks very similar to the above structure, but the number of parameters of the model increases with the maximum number of operations.

**Fully-connected.** It has 128 and 256 hidden units in each of the two fully connected layers. Like CNN, there is a waste of computational graphs. In addition, because there is no weight sharing, it is not suitable for the problem of choosing the best job among candidate jobs. In addition, as with CNN parameters, the parameters of the structure increase with the maximum number of jobs.

**RankSVM.** It is also applicable to the problem of selecting only the most suitable candidate with a special algorithm for determining the rank of candidates. However, due to the nature of SVM, it takes a lot of time to process a lot of data at once, and it is not free from parameter tuning and feature extraction because it is necessary to select the kernel well in the process of fitting complex functions including nonlinear functions.

Table 2a presents the results of various algorithmic structures. They use all raw features in Table 1. The number next to the algorithm(i.e. CNN 128, Fc 1) in the first column of the table indicates the number of jobs in one mini-batch and Fc means Fully-connected. In Table 2b, we multiply the features that is the criterion of dispatching rule selection according to the situation to the raw feature in Table 2a, and the Table 2b presents the result using it as a feature. In both Table 2a and 2b, VLCNN has a much higher accuracy than other architectures. This is because, as mentioned before, VLCNN can cope with variable input and output unlike other neural networks. And also as one of the general advantages of neural network, it can fit complex dispatching rules well. This experiment considered a case where a virtual human expert is involved in a virtual factory simulator, but it is quite promising to be applied in actual manufacturing factories because it achieves a much higher level of accuracy than other methodologies including the rich experience of human expert perfectly. Since the network used in VLCNN itself is flexible, so it is possible to express and learn more complex rule by adding sophisticated and outbreaking neural network.

From a computational point of view, unlike other neural networks where the number of parameters increases with the maximum number of candidates, VLCNN can have a constant number of parameters by fully sharing its weights. Also with DCG, using the internal computation method similar to the existing neural network, it is possible to compute quickly using the GPU.

Table 2: Accuracy on Two Feature sets

| (a) Raw Feature | | | (b) Product Feature | | |
|---|---|---|---|---|---|
| **Architecture** | **Training** | **Testing** | **Architecture** | **Training** | **Testing** |
| VLCNN | 96.34% | 95.27% | VLCNN | 99.46% | 99.37% |
| RankSVM | 92.34% | 91.33% | RankSVM | 94.19% | 93.94% |
| CNN 1 | 99.32% | 53.32% | CNN 1 | 99.43% | 57.85% |
| CNN 128 | 77.88% | 50.03% | CNN 128 | 86.58% | 54.13% |
| Fc 1 | 68.86% | 40.76% | Fc 1 | 77.68% | 40.97% |
| Fc 128 | 42.13% | 36.10% | Fc 128 | 44.05% | 35.14% |

## 5 CONCLUSION

In this paper, we newly define "Learning to Select" problem and suggest VLCNN to solve the problem. It combines the power of classification in neural network with the variability from LtoR algorithm. For the job selection experiment, it shows better performance than other well-known

algorithms. It also shows the computational benefit by reducing the candidates before computation. There are many studies that can be done. It is worthwhile to apply the same job selection problem to real factory data. This is because it will discover various new problems and development directions of the suggested algorithm during applying real data, as well as the value of the research in the manufacturing industry is very high.

Also, there are many other domains that can be used. Selecting the best representing object at each frame for video is one of challenging domain that can be applied. Because we do not know how many objects are detected every moment, it is hard to fix the input and output size of the algorithm for selecting the best of them. For example, in order to caption a video or explain the situation of a video, it is necessary to detect objects at every frame in the video and to select the object that best describes the current frame. At this time, a different number of objects are detected every frame, and the object corresponding to the current frame must be selected. However, it is difficult to select an object by fixing the structure if the number of objects that can be selected each time is changed even in the case of a neural network that is most representative in image or image classification.

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

## APPENDIX

### LEARNING CURVE FOR EXPERIMENT

Figure 3a and 3b show test accuracy for each epoch when learning data with raw feature and product feature using VLCNN. The accuracy for five different random seeds is shown, with the solid black line representing the average and the gray area representing mean $\pm$ standard deviation. In both cases, it is confirmed that the learning curve is smooth and the test accuracy is robust.

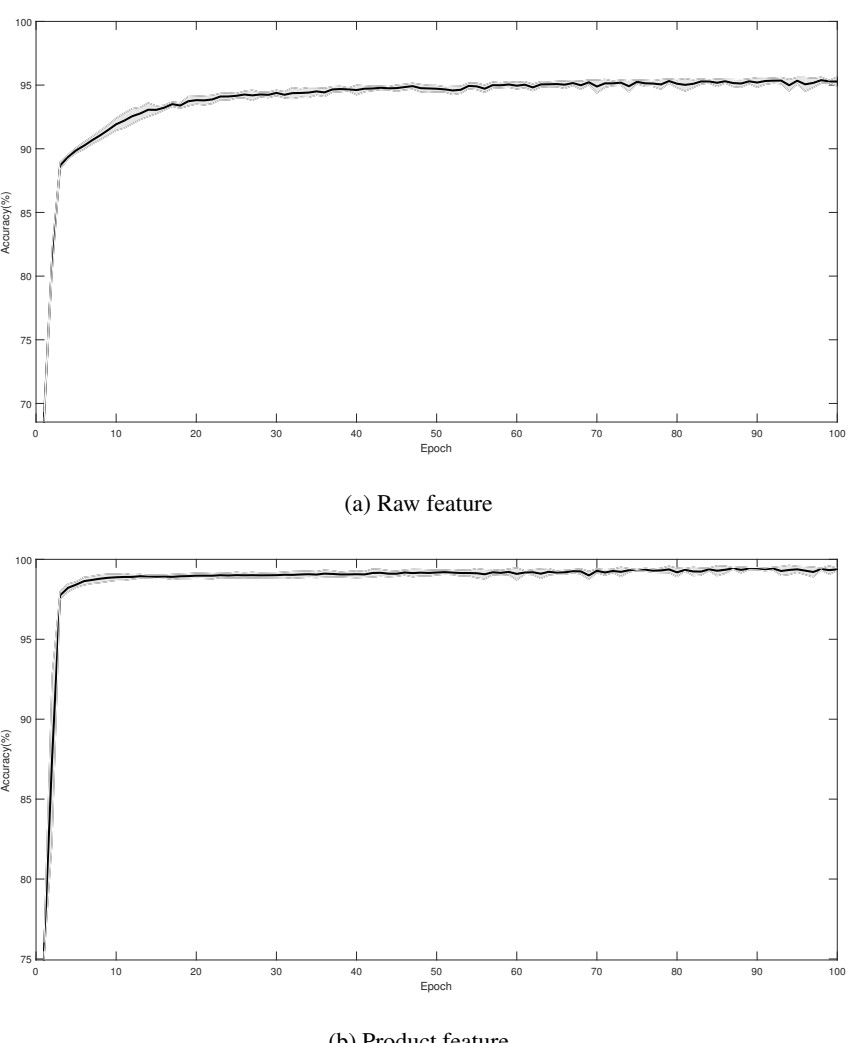

(a) Raw feature

(b) Product feature

Figure 3: Learning curve of VLCNN

