# OpenReview forum: "Learning to Select: Problem, Solution, and Applications"
_ICLR.cc/2018/Conference — Reject_

### Official Review · AnonReviewer3 · 2017-11-27
**The paper proposed a new framework called `Learning to select’, in which a best candidate needs to be identified in the decision making process such as job dispatching. A CNN architecture is designed, called `Variable-Length CNN’, to solve this problem.**

**Rating:** 4
**Confidence:** 4

**Review:**

The paper proposed a new framework called `Learning to select’, in which a best candidate needs to be identified in the decision making process such as job dispatching. A CNN architecture is designed, called `Variable-Length CNN’, to solve this problem.

My major concern is on the definition of the proposed concept of `learning-to-select’. Essentially, I’ve not seen its key difference from the classification problem. While `even in the case of completely identical candidates, the label can be 1 in some situations, and in some other situations the label can be 0’, why not including such `situations’ into your feature vector (i.e., x)? Once you do it, the gap between learning to select and classification will vanish. If this is not doable, you should better make more discussions, especially on what the so-called `situations’ are.  Furthermore, the application scope of the proposed framework is not very well discussed. If it is restricted to job dispatching scenarios, why do we need a new concept “learning to select”?

The proposed model looks quite straightforward. Standard CNN is able to capture the variable length input as is done in many NLP tasks. Dynamic computational graph is not new either. In this sense, the technical novelty of this work is somehow limited.

The experiments are weak in that the data are simulated and the baselines are not strong. I’ve not gained enough insights on why the proposed model could outperform the alternative approaches. More discussions and case studies are sorely needed.

---

### Official Review · AnonReviewer2 · 2017-11-30
**An application of Dynamic Computation Graph with CNN structure on job dispatching data**

**Rating:** 4
**Confidence:** 4

**Review:**

This paper proposes a "Learning to Select" problem which essentially is to select the best among a flexible size of candidates, which is in fact Learning to Rank with number of items to select as 1. To be able to efficiently train the model without wasting time on the items that are not candidates, the authors applied an existing work in literature named Dynamic Computation Graph and added convolutional layer, and showed that this model outperforms baseline methods such as CNN, fully-connected, Rank-SVM etc.

As this paper looks to me as an simple application of an existing approach in literature to a real-world problem, novelty is the main concern here. Other concerns include:
1. Section 2. It would be good to include more details of DCG to make the papers more complete and easier to read.
2. It looks to me that the data used in experimental section is simulated data, rather than real data.
3. It looks to me that baselines such as CNN did not perform well, mainly because in test stage, some candidates that CNN picked as the best do not actually qualify. However, this should be able to be fixed easily by picking the best candidate that qualify. Otherwise I feel it is an unfair comparison to the proposed method.

---

### Official Review · AnonReviewer1 · 2017-12-08
**The paper proposes a method to study the learning to select problem using a variable length CNN**

**Rating:** 4
**Confidence:** 5

**Review:**

The authors state
"This problem is simply a matter of choosing the best candidate among the given candidates in a specific situation." -> It would be nice to have examples of what constitutes a "candidate" or "situation"?

The problem definition and proposed approach needs to be made more precise. For e.g. statements like "
For the problem of classifying data into several classes, neural networks have dominated with high performance. However, the neural networks are not an adequate answer for every problem." are not backed with enough evidence.

A detailed section on related work would be beneficial.

---

### Decision · Program_Chairs · 2018-01-29
**ICLR 2018 Conference Acceptance Decision**

**Decision:**

Reject

**Comment:**

Three reviewers recommended rejection and there was no rebuttal.